# Green synthesis of stable hybrid biocatalyst using a hydrogen-bonded, π-π-stacking supramolecular assembly for electrochemical immunosensor

Wei Huang[1,2], Haitao Yuan[3], Huangsheng Yang[2], Xiaomin Ma [4], Shuyao Huang[5], Hongjie Zhang[2], Siming Huang [6], Guosheng Chen [2] ✉ & Gangfeng Ouyang [1] ✉

Rational integration of native enzymes and nanoscaffold is an efficient means to access robust biocatalyst, yet remains on-going challenges due to the trade-off between fragile enzymes and harsh assembling conditions. Here, we report a supramolecular strategy enabling the in situ fusion of fragile enzymes into a robust porous crystal. A c2-symmetric pyrene tecton with four formic acid arms is utilized as the building block to engineer this hybrid biocatalyst. The decorated formic acid arms afford the pyrene tectons high dispersibility in minute amount of organic solvent, and permit the hydrogen-bonded linkage of discrete pyrene tectons to an extended supramolecular network around an enzyme in almost organic solvent-free aqueous solution. This hybrid biocatalyst is covered by long-range ordered pore channels, which can serve as the gating to sieve the catalytic substrate and thus enhance the biocatalytic selectivity. Given the structural integration, a supramolecular biocatalyst-based electrochemical immunosensor is developed, enabling the pg/mL detection of cancer biomarker.

Enzymes is a class of natural biocatalysts initiating countless life-sustaining biotransformation in living system, featuring high efficiency unmatched by artificial catalysts[1–4]. The majority of enzymes are workable in biocompatible conditions, making enzymatic catalysis energy-efficient and environmentally friendly. Considering these attributes, substantial efforts have been devoted to control over enzymes in synthetic chemistry, environmental remediation, energy harvest and biosensor designs over the past decades[5,6]. The flexible structures of enzymes, however, are highly susceptive and easy to be denatured under external stimulus, which greatly limit their large-scale industrial applications[5].

Tightly confining native enzyme using a nanoscaffold is an efficient means to design hybrid biocatalysts, which is able to solve the stability problem and enhance the recoverability of an enzyme. In this regards, porous organic frameworks, such as metal-organic frameworks (MOFs) and covalent-organic frameworks (COFs), are the ideal

[1]School of Chemical Engineering and Technology, Sun Yat-sen University, 519082 Zhuhai, China. [2]MOE Key Laboratory of Bioinorganic and Synthetic Chemistry, School of Chemistry, Sun Yat-sen University, 510275 Guangzhou, China. [3]Department of Geriatric Medicine, Shenzhen People's Hospital (The Second Clinical Medical College, Jinan University), 518020 Shenzhen, China. [4]Cryo-EM Center, Southern University of Science and Technology, 518055 Shenzhen, China. [5]Instrumental Analysis and Research Center, Sun Yat-sen University, 510275 Guangzhou, China. [6]Guangzhou Municipal and Guangdong Provincial Key Laboratory of Molecular Target & Clinical Pharmacology, the NMPA and State Key Laboratory of Respiratory Disease, School of Pharmaceutical Sciences and the Fifth Affiliated Hospital, Guangzhou Medical University, 511436 Guangzhou, China. ✉e-mail: chengsh39@mail.sysu.edu.cn; cesoygf@mail.sysu.edu.cn

nanoplatform to target this goal when taking these into account: (1) the ultrahigh porosity offers high enzyme loading efficiency; (2) the unambiguous topologies of these frameworks allow the in-depth understanding of the catalytic mechanism. Indeed, scientists have developed several in situ encapsulation strategies to synthesize MOF- or COF-confined biocatalysts[7]. This bottom-up approach is convenient and efficient, without the need of complicated pore engineering[8–17]. However, it still remains technical challenges, mainly originating from the harsh crystallization conditions to which enzyme is susceptive. For MOFs, the nanoscaffold suitable for enzyme encapsulation, to date, were limited at the zeolitic imidazolate frameworks (ZIFs) family (ZIF-8, ZIF-90 and metal azolate framework-7, etc.)[18–21]. Unfortunately, ZIFs have narrow pore aperture (ca. 3.4 Å), which significantly reduces the accessibility of the interior enzymes[22,23]. While for COFs, it tends to generate an amorphous or low crystalline phase under biologically friendly environments, and the enormously ambiguous domain limits the applications of COF hybrid biocatalyst[24].

Hydrogen bond-mediated supramolecular crystals, such as hydrogen-bonded organic frameworks (HOFs), are a rising class of porous frameworks complimentary to MOFs and COFs. This porous crystal is orderly linked by discrete molecular tectons with inter-molecular hydrogen bonds, featuring the merits of mild crystallization condition, high solvent processability, metal-free biocompatibility, etc[25–27]. Very recently, scientists have demonstrated the possibility of encapsulating enzymes using a charge-assisted, hydrogen-bonded supramolecular framework[28]. However, the narrow pore aperture (ca. 6.4 Å) of this supramolecular framework still limited the mass transfer and the accessibility of the interior enzymes. Beyond that, the strength of this charge-assisted H-bonded interaction is still weak, which makes the structure stable only in a pH range of 5–10. Our group showcases the feasibility of using pyrene modules to strengthen the enzyme-HOF voxel[29–32]; however, it requires milliliters of organic solvent to disperse the pyrene modules and thus induces the supramolecular assembly. This organic solvent dosage may limit the practicability of this strategy to some fragile enzymes. As far as we known, the green and bio-compatible supramolecular approach[33,34] able to engineer hybrid bio-catalyst with highly structural stability and bioactivity is seldom reported.

Herein, we report a green and almost organic solvent-free supramolecular assembly approach to synthesize stable and efficient HOF hybrid biocatalyst, in which a pyrene tecton with four formic acid arms (1,3,6,8-tetracarboxy pyrene, $H_4$TCPy) is used as the supramole-cular building block (Fig. 1). The decorated formic acid arms afforded this pyrene tectons higher dispersibility in minute amount of solvent, and then permitted the hydrogen-bonded linkage of discrete pyrene tectons to an extended supramolecular network around an enzyme in almost organic solvent-free aqueous solution. While the coplanar pyrene core offered a strong layer-by-layer π-π-stacking interaction, strengthening the HOF hybrid biocatalyst in a wide range pH from 1 to 11. Using this in situ approach, a series of enzymes with different sur-face chemistries were readily assembled into a highly crystalline supramolecular biocatalyst. Importantly, the crystallographic struc-ture of this supramolecular biocatalyst as well as its ordered pore channels were unambiguously identified by the advanced low-dosage-electron cryo-electron microscopy (cryo-EM) technique. This hybrid biocatalyst showed higher catalytic ability compared to free enzyme in non-physiological environment, and its explicit pore channel could serve as the gating to sieve the catalytic substrate and thus enhance the biocatalytic selectivity. By means of the structural integration, a HOF electrochemical immunosensor was reported, which enabled the ultrasensitive detection of cancer biomarker, mucin-1 (MUC1).

## Results
### Preparation and structural characterization of HOF hybrid biocatalysts

Horseradish peroxidase (HRP) is an enzyme widely used in clinical assay, contamination remediation and catalytic therapy, etc[35]. We took HRP as the model enzyme to verify this supramolecular strategy. In this supramolecular strategy, it does not require any energy-intensive steps and the organic solvent consumption is very limited. When 9 mL of HRP aqueous solution (0.54 mg/mL) was added into 300 μL of well-dispersive $H_4$TCPy dimethyl formamide (DMF) solution, a mass of biocomposites were generated (Supplementary Fig. 1). Notably, the sediment still could be formed when replacing HRP aqueous solution by equivalent water (Supplementary Fig. 2), implying that the enzyme might be fused through a co-precipitation process[36]. After washing the

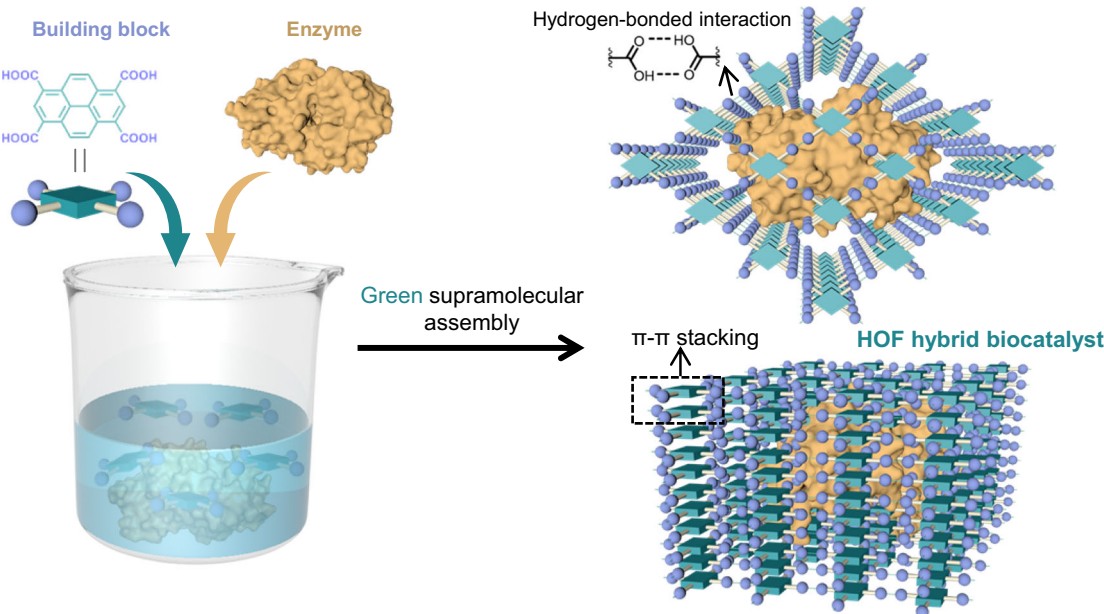

**Fig. 1 | Schematic presentation of the green supramolecular assembly strategy.** A green and almost organic solvent-free supramolecular assembly approach to synthesize stable and efficient HOF hybrid biocatalyst. The pyrene core and four formic acid arms in the building block are shown as dark green prismatic square and purple ball, respectively.

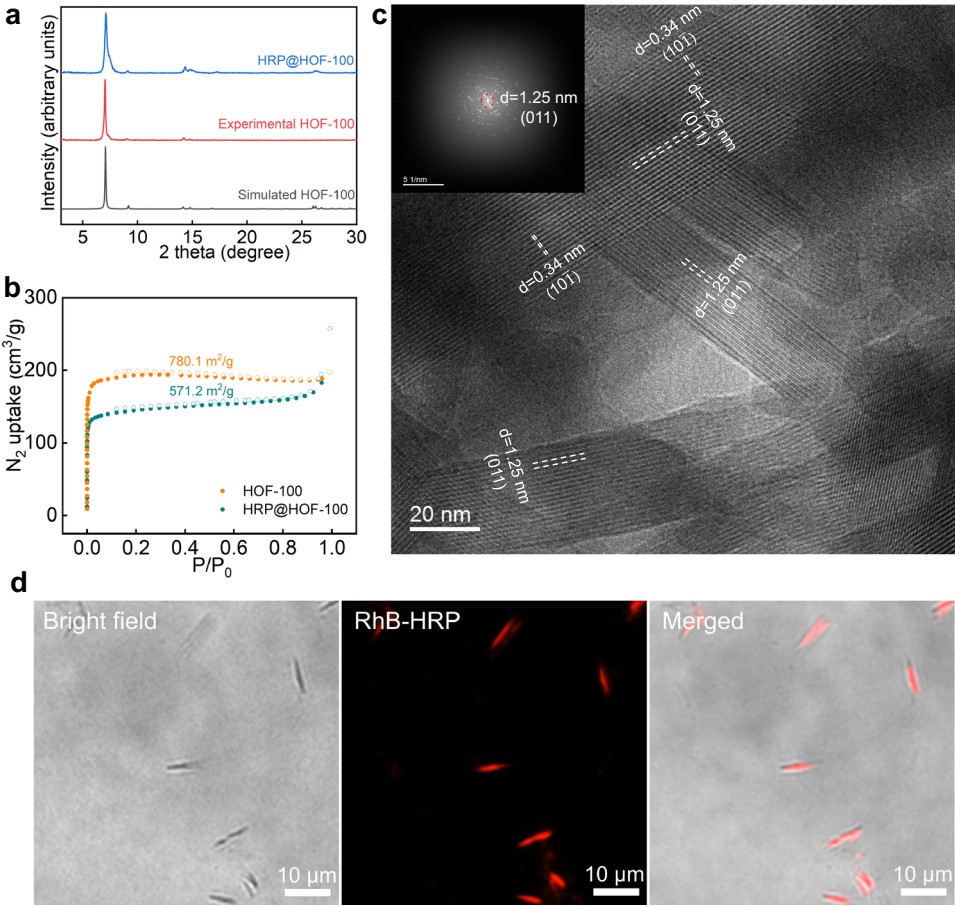

**Fig. 2 | Preparation and structural characterization of HOF hybrid biocatalysts.** **a** PXRD patterns of HRP@HOF-100, experimental HOF-100 and simulated HOF-100. **b** Nitrogen adsorption/desorption isotherm of HRP@HOF-100 and pure HOF-100. **c** The cryo-EM image showing the long-range ordered channels within HRP@HOF-100. The inset was the fast Fourier transform image. **d** The CLSM images showing the spatial distribution of HRP (HRP was labeled by red dye) within HOF-100 scaffold.

sediment by deionized water 2 times and ethanol 1 time and drying, the crystallographic structure of the as-synthesized biocomposite was examined by powder X-ray diffraction (PXRD), and it showed strong Bragg diffraction pattern (Fig. 2a), matching well with that of the standard HOF-100 (a recrystallized product of $H_4TCPy$[37], Supplementary Fig. 3). Viewing from the scanning electron microscopy (SEM) image, this HOF biocomposite (denoted as HRP@HOF-100) presented a rod-shaped microcrystalline structure (Supplementary Fig. 4). These results suggested that the nanocrystalline HOF biocomposites were indeed formed under the green and biocompatible environment.

The enzyme loading efficiency was surveyed through the measurement of the concentration differences of the supernatants before and after assembly using standard Bradford proteins assay (Supplementary Fig. 5), and it gave a ca. 31.8% wt HRP loading (Supplementary Table 2), which outperformed the most of reported methods using porous organic frameworks (Supplementary Table 3). Next, we sought to verify the encapsulation of HRP within the HOF-100 scaffold. In Fourier-transform infrared (FTIR) spectra (Supplementary Fig. 6), the characteristic peak at 1700–1610 cm$^{-1}$, regarded as the typical amide I band of an enzyme[38], was observed in HRP@HOF-100, but not in standard HOF-100. This was the evidence of the fusion of HRP. In addition, the fusion of HRP was also endorsed by the thermogravimetric analysis (TGA), wherein the weight loss at ca. 200–350 °C was caused by the pyrolysis of proteins (Supplementary Fig. 7). The insight into the porous structure was further examined by nitrogen adsorption/desorption isotherm experiment (Fig. 2b). The typical I-type curves revealed the microporous structure of the as-synthesized HRP@HOF-100, with a

calculated Brunner–Emmet–Teller (BET) surface area as high as 571.2 m$^2$/g. Nevertheless, this BET value was reduced compared to the pure HOF-100 (780.1 m$^2$/g), manifesting that the interior pore of HOF-100 was partially occupied by HRP. Furthermore, the pore-size distributions (Supplementary Fig. 8), calculated by nonlocal density functional theory (NLDFT) method, showed that both the micropores of HOF-100 and HRP@HOF-100 centered at 0.73 nm and 1.2 nm, which were in line with the theoretical pore dimension (0.8 nm × 1.2 nm). Compared to HOF-100, HRP@HOF-100 showed a deceased pore volume, and this was caused by the spatial occupation by the enzymes. Concurrently, we also carried out an enzyme-adsorption experiment to elucidate that the bulky HRP (ca. 4.0 nm × 4.4 nm × 6.4 nm) was unable to infiltrate into the micropores of HOF-100 or surface-adsorb onto HOF-100 (detail seen in Supplementary Fig. 9), evidencing that the HRP was indeed fused into HOF-100 during this green bottom-up assembly.

To visualize the spatial distribution of HRP, we performed the confocal laser scanning microscope (CLSM) experiment, in which HRP was pre-labeled with the red dye, rhodamine B (RhB). As demonstrated in Fig. 2d, the red fluorescence overlapped perfectly with the HOF-100 nanoscaffold, confirming that the fused HRP was uniformly confined within the framework. Consequently, all the aforementioned results verified that this in situ HOF approach permitted the high-efficient enzyme encapsulation under biocompatible environment.

The structure of the nanoscaffold significantly influences the bioactivity of an encapsulated enzyme. We attempted to visualize the microstructure of the HOF hybrid biocatalyst using low-electron-dose cryo-electron microscopy (cryo-EM) technique (dose rate: ~15 counts/

pixel/second; total exposure time: 0.23 s; total dose rate: ca. 30 e⁻/Å² per micrograph). As displayed in Fig. 2c and Supplementary Figs. 10–13, the long-range ordered channels along the (011) lattice plane were clearly witnessed through the HRP@HOF-100 biocatalyst, and the channel width was identified to be ca. 1.2 nm (Fig. 2c). In addition, the fast Fourier transform (FFT) image rendered various diffraction spots from different lattice plane (Supplementary Figs. 10–13), indicating the high crystallinity of the HRP@HOF-100 biocatalyst. Particularly, the (10Ī) lattice plane, ascribed to the layer-by-layer π-π stacking, was measured to be ca. 0.34 nm, indicating a strong π-π stacking force (Fig. 2c and Supplementary Fig. 10b). Notably, the 1.2 nm width channel of this HOF hybrid biocatalyst was larger than the ones of the reported enzyme@bioHOF-1 (ca. 0.64 nm)[27], and might favor the biocatalytic power of an encapsulated enzyme.

## Enzyme conformation study

We then attempted to study the enzymatic conformation in this HOF hybrid biocatalyst. HRP is a heme protein, which has a five coordinate, high-spin heme center (Fig. 3a). This in situ supramolecular assembly approach was carried out in a ca. 3% (v/v) DMF aqueous solution (9 mL deionized water + 300 μL DMF), and the whole procedure was completed within 30 min at room temperature. We confirmed that this assembling condition was biocompatible to the fragile enzyme. When HRP was directly incubated in 3% (v/v) DMF aqueous solution at room temperature, the conformation of enzyme could be well preserved even after 6 h, as evidenced by the UV-Vis spectra (Supplementary Fig. 14a) and circular dichroism (CD) spectra (Fig. 3b and Supplementary Table 7). In addition, the catalytic activity was well retained after this exposure experiment (Supplementary Fig. 14b).

The biointerface between HRP and HOF in the as-synthesized HRP@HOF-100 biocatalyst was examined by solid-state nuclear magnetic resonance (ssNMR, Fig. 3c). The chemical shift at 12.8 ppm in H¹ ssNMR of HRP@HOF-100 biocatalyst is assigned to the proton of the carboxyl groups of H₄TCPy molecular tectons. This peak was observed to be shifted into a low magnetic field compared to the physical mixed sample of HRP and HOF-100 ($\delta = 11.1$ ppm), suggesting the biointerface

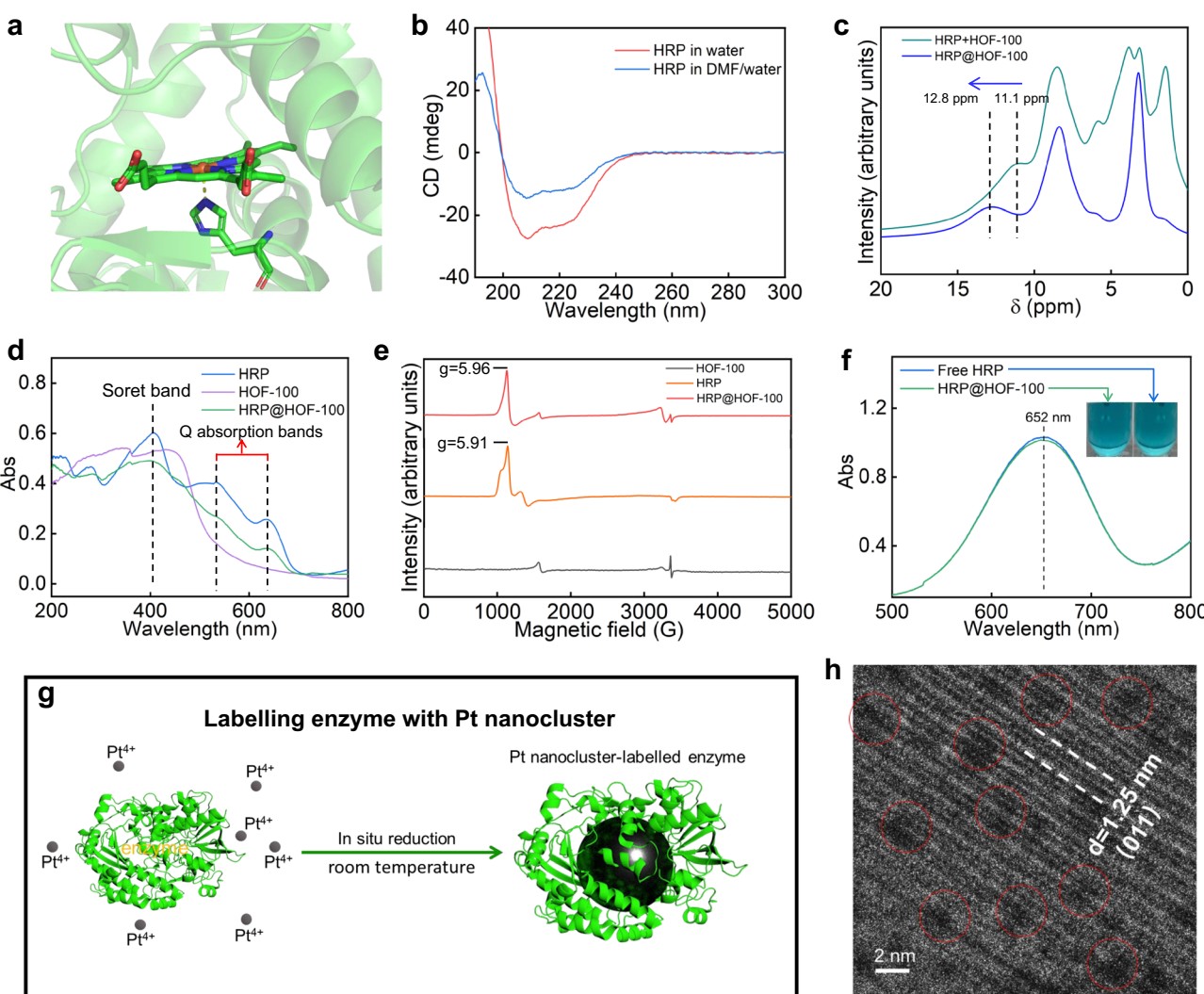

**Fig. 3 | Enzymatic conformation study. a** The microenvironment of heme center of native HRP (adapted from PDB: 1hch). The colors used in heme center of HRP are: green for C atom; blue for N atom; red for O atom; the central Fe ion is highlighted as an orange ball. **b** The CD spectra of free HRP in water and HRP after incubating in DMF aqueous solution for 6 h. The volumes of DMF and deionized water were 0.3 mL and 9.0 mL, respectively, which was in agreement with the assembly system. **c** The H¹ ssNMR spectra of HRP@HOF-100 biocomposite and its physical mixed sample of HRP and HOF-100. **d** UV-Vis DRS of native HRP, HOF-100 and HRP@HOF-100. **e** The 0–5000 G region of the EPR spectra of native HRP, HOF-100 and HRP@HOF-100. **f** The UV-Vis spectra for free HRP and HRP@HOF-100 after 5 min reaction time. Inset is the photograph of corresponding output signal. Conditions: 0.1 mM H₂O₂. The concentrations of free HRP and HRP@HOF-100 (equivalent immobilized HRP) were 0.125 mg/mL. **g** Schematic representation of the Pt NCs labeling process. The structure of GOx is adapted from PDB: 1 gal. **h** High-magnification cryo-EM image of GOx-Pt NCs-encapsulated HOF-100. The mono-dispersed Pt NCs were highlighted in red circles.

interaction between HRP and the carboxyl groups of HOF-100 scaffold[30]. To verify whether such biointerface could affect the active center of HRP, the UV-Vis diffuse reflectance spectra (UV-Vis DRS, Fig. 3d) experiment was implemented. The absorption profile of UV-Vis DRS is highly sensitive to the structure change of the active center of heme protein[39]. The unchanged Soret bands (~408 nm) and Q absorption bands (500 nm to 700 nm) between native HRP and HRP@HOF-100 biocatalyst manifested that this green supramolecular assembly had a negligible effect on the active center of HRP[40]. Electron paramagnetic resonance (EPR) spectra were further employed to examine the coordinative environment of the HRP heme center (Fig. 3e). In the native HRP sample, there is a high-spin Fe component with a g-value of 5.91, which is the characteristic of five coordinate heme species[41]. We found that the high-spin EPR component of HRP@HOF-100, featuring the g-value at 5.96, was consistent with that in the native HRP, evidencing that the five coordinate, high-spin heme conformation was maintained in the HRP@HOF-100 biocatalyst. These conformational insights by spectroscopic methods confirmed the reliability of this HOF supramolecular strategy for engineering hybrid biocatalysts. We next evaluated the catalytic capacity of this HRP@HOF-100 biocatalyst using 3,3',5,5'-tetramethylbenzidine (TMB) as the peroxidase substrate. After 5 min reaction time, the catalytic product of oxidized TMB (Abs at 652 nm) by HRP@HOF-100 biocatalyst was comparable to that by free HRP under identical enzyme dosage (Fig. 3f), attributing to the high permeability of the porous HOF-100 scaffold.

In order to locate the enzyme within HOF-100, we attempted to label the enzyme with nanocluster of heavy element, wherein the Pt nanoclusters (Pt NCs) were in situ reduced within an enzyme using a biocompatible method[42] (Fig. 3g; glucose oxidase, GOx, was used herein because of the ease of labeling). This method afforded the Pt NCs-labeled enzyme yet had limited influence on the conformation of an enzyme (Supplementary Fig. 15 and Table 8). As shown in Supplementary Fig. 16a, the as-synthesized Pt NCs with ca. 2.5 nm dimension were monodisperse, attributing to the stabilization effect by enzyme. We further confirmed the successful labeling of Pt NCs by a protein negative staining experiment (detail seen in Supplementary Method). Under the protein negative staining experiment, we clearly observed the enzyme outline (in white) around individual Pt NCs, suggesting that each enzyme was successfully labeled by a Pt NC (Supplementary Fig. 16b). In addition, the insight into the molecular weight by matrix-assisted laser desorption ionization time-of-flight mass spectrometry (MALDI-TOF MS) showed that the enzyme weight increased from 76794 to 78606 after Pt NCs labeling (Supplementary Fig. 17), and it gave a labeling efficiency of ca. nine Pt per one enzyme. These results adequately demonstrated the highly efficient labeling of Pt NCs in enzyme. Under the cryo-EM, we got a glimpse of the spatial location of enzyme within HOF-100. The Pt NCs-labeled enzymes (GOx-Pt NCs) were discretely located in HOF-100 scaffold (Supplementary Fig. 18). In addition, in the high-magnification cryo-EM imaging, we identified the highly crystalline HOF-100 pores and many monodispersed GOx-Pt NCs (highlighted in red circles) simultaneously (Fig. 3h). Based on this labeling experiment, we inferred that the encapsulated enzymes were discretely located in HOF-100, which favored the biocatalysis in a confined microenvironment.

## The extendibility of this supramolecular strategy

The possibility of this green supramolecular strategy for other proteins was surveyed. We chose several proteins with various surface chemistries, including bovine serum albumin (BSA), cytochrome c (Cyt c), glucose oxidase (GOx), myoglobin (MB), trypsin and urease to verify the extendibility of this strategy. The molecular weights and surface charges of these proteins were summarized in Supplementary Table 1. All the as-synthesized protein biocomposites displayed the typical PXRD patterns of standard HOF-100, indicating the highly crystallinity of proteins@HOF-100 (Supplementary Fig. 19). Likewise, all the

proteins@HOF-100 presented rod-like nanostructures (Supplementary Fig. 20). The insight into the protein encapsulation was examined by FT-IR spectra, wherein the emerging absorbance bands around 1700-1610 cm$^{-1}$, assigned to the typical amide I bands of the proteins[38], were recorded in all these proteins@HOF-100 (Supplementary Fig. 21). In addition, the successful encapsulation of proteins within HOF-100 scaffold was further supported by the TGA results, wherein the weight loss at ca. 200-350 °C was caused by the pyrolysis of encapsulated proteins (Supplementary Fig. 22). The protein loading efficiencies, evaluated by standard Bradford protein assay, were calculated to be 33.3% wt to 38.0% wt (Supplementary Table 2), also outperforming the most of reported methods (Supplementary Table 3). These results implied the general applicability of this HOF supramolecular strategy for synthesizing biohybrid materials.

We further studied the catalytic kinetics of the enzymes, including Cyt c and GOx after the fusion by HOF-100. The conformation profiles by UV-Vis spectra (Supplementary Fig. 23), CD spectra (Supplementary Figs. 24 and 26 and Supplementary Tables 9 and 10) and fluorescence spectra (Supplementary Fig. 25) suggested that both Cyt c and GOx preserved their native conformations under the aqueous solution of HOF assembly required. As a result, the prepared Cyt c@HOF-100 (Supplementary Fig. 27) and GOx@HOF-100 biocatalysts (Supplementary Fig. 28) showed comparable catalytic rates to their free counterparts under a series of concentrations of substrates. Such well-preserved activity of enzyme@HOF-100 biocatalysts were attributed to: (1) the green and biocompatible HOF growth environment could minimize the disturbance of enzyme's conformation; and (2) the periodically arranged pore channels with large opening window promoted the mass transfer, favoring the biocatalysis in a confined environment.

## HOF gating effect for enhancing the biocatalytic selectivity

In nature, a large number of enzymes have relatively poor substrate specificity, which are closely related to the stereoscopic conformation of the active sites. For instance, Lipase is an important hydrolase that widely used for the hydrolysis of esters in industry[43,44], however, the poor substrate specificity limits its application in selective catalysis. We next aimed to demonstrate that our biocatalyst could solve this problem through the gating effect of HOF. In this regard, a Lipase (from candida rugose) @HOF-100 biocatalyst was constructed by the similar HOF supramolecular assembly, and the detailed structural analyses were provided in Supplementary Figs. 29–39. Briefly, both the FT-IR (Supplementary Fig. 32), TGA (Supplementary Fig. 33) and CLSM experiments (Supplementary Fig. 34) confirmed the successful incorporation of Lipase. In addition, the CD spectrum showed that the conformation of Lipase could be well preserved in this mild assembling condition (Supplementary Fig. 38). The insight into the biointerface was examined by ssNMR (Supplementary Fig. 39), in which the chemical shift at 12.7 ppm in H$^1$ ssNMR of Lipase@HOF-100 biocatalyst was assigned to the proton of the carboxyl groups of H$_4$TCPy molecular tectons. This peak was observed to be shifted into a low magnetic field compared with the physical mixed sample of Lipase and HOF-100 ($\delta$ = 11.5 ppm), suggesting the interaction between Lipase and the carboxyl groups of HOF-100 scaffold.

Viewing from the nano-architecture of the Lipase@HOF-100 biocatalyst, the highly crystalline HOF scaffold can sever as the gatekeeper to sieve the biocatalytic substrate, and this gating effect is relied on the geometric dimension of the 1D channel window. We attempted to identify the geometry of the channel window utilizing cryo-EM. Under the [10$\bar{1}$] project, the rhombus-shaped channel windows with 1.2 nm × 0.8 nm dimension were clearly captured under a low-electron-dose imaging model (Fig. 4a). These uninterrupted and unambiguous channel windows were formed by the intermolecular hydrogen-boned linkages of four H$_4$TCPy tectons, which were in lined with the theoretical structure (Fig. 4a).

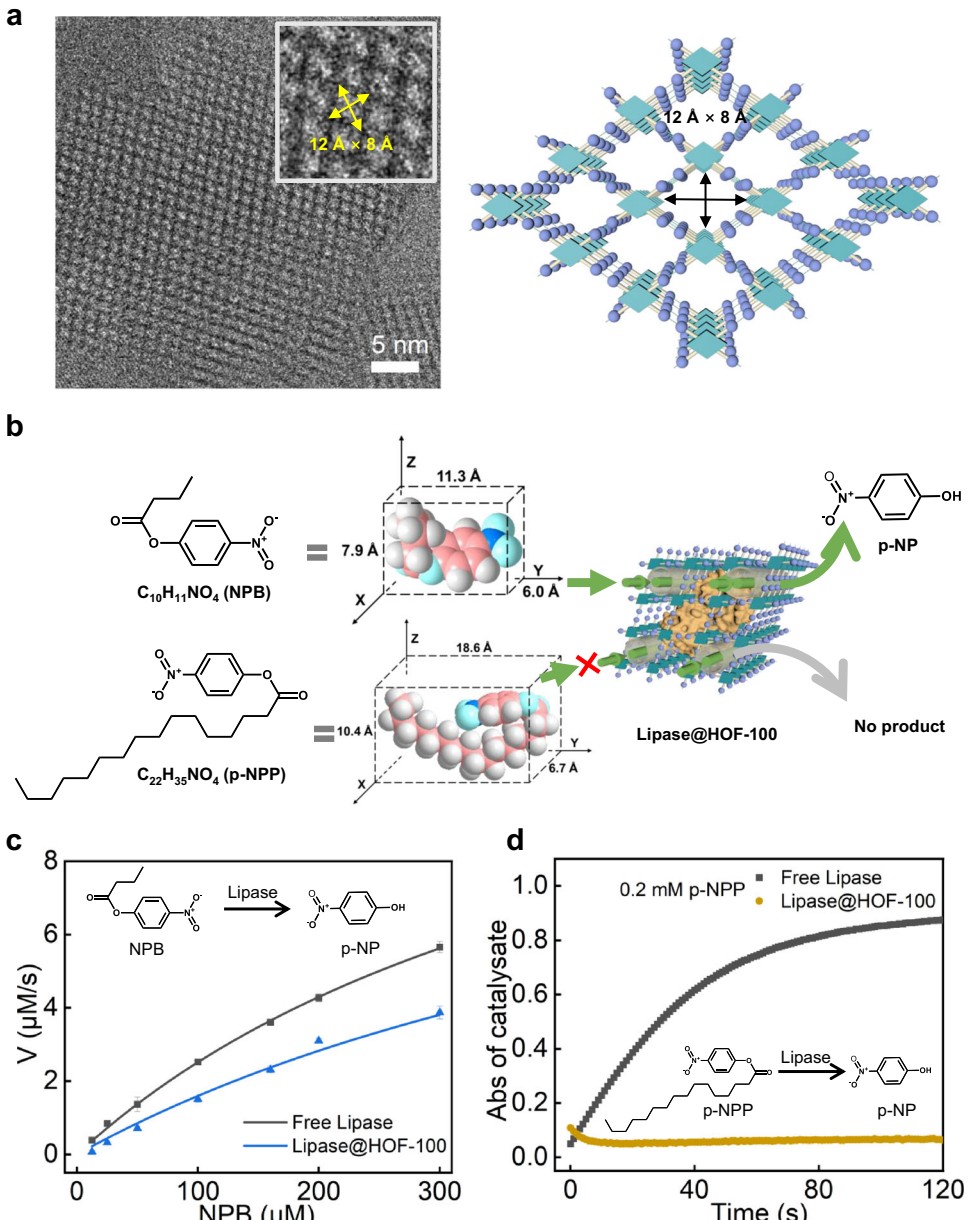

**Fig. 4 | HOF Gating effect for enhancing the biocatalytic selectivity. a** The cryo-EM image identifying the rhombus-shaped pore windows from the [10$\bar{1}$] projection, and the inset is the enlarged domain. The pyrene core and formic acid arm in HOF structure are shown as dark green prismatic square and purple ball, respectively. **b** The schematic illustration of the gating effect of Lipase@HOF-100 for ester hydrolysis. The color used in NPB and p-NPP models are: while ball for H atom; pink ball for C atom; blue ball for N atom; cyan ball for O atom. While the pyrene core, formic acid arm and Lipase in Lipase@HOF-100 structure are shown as dark green

prismatic square, purple ball and yellow surface model, respectively. **c** Plot of reaction velocity, V, against substrate [NPB]. The dosages of for Lipase and Lipase@HOF-100 were kept at the same in each trial (80 μg/mL). The data of 3 independent experiments and calculated error bars (SD) are presented in c, SD = Standard Deviation (n = 3). Data are presented as mean values +/−SD. **d** The catalytic kinetics curves of free Lipase and Lipase@HOF-100 under 0.2 mM p-NPP. The dosages of Lipase (80 μg/mL) used were kept at the same in each trial.

The short-chain ester, p-nitrophenyl butyrate (NPB, ca. 6.0 Å ×7.9 Å ×11.3 Å molecular dimension, Fig. 4b), was chosen as the substrate. Lipase can hydrolyze NPB into p-nitrophenyl (p-NP), which can be quantified based on the absorbance at 405 nm (Supplementary Fig. 40). When the Lipase@HOF-100 biocatalyst was used to hydrolyze NPB, it showed a comparable hydrolysis kinetics to the native Lipase (Supplementary Fig. 41), because of the high permeability of the 1D channel towards NPB. In this case, the enzymatic Michaelis-Menten parameters were carefully investigated. Figure 4c described the substrate concentration-dependent catalytic rates for Lipase@HOF-100 and free Lipase under identical enzyme dosage. We found that all the calculated catalytic rates of Lipase@HOF-100 were close to the free

counterpart. In addition, Lipase@HOF-100 held comparable catalytic kinetic parameters including maximum catalytic rate ($V_{max}$) and Michaelis constant ($K_m$) to the free Lipase (Supplementary Table 4). It should be pointed out that, when enzymes were encapsulated into ZIF-8, a mostly widely used MOFs scaffold with 0.34 nm pore window, the catalytic performances were found to be inhibited in most cases[12,45].

When a long-chain ester of p-nitrophenyl palmitate (p-NPP, 6.7 Å × 10.4 Å × 18.6 Å, Fig. 4b) was chosen as the catalytic substrate, the native Lipase also could efficiently hydrolyze it because of the poorly stereoscopic selectivity of the active site of Lipase. Conversely, the slopes of catalytic kinetic curves of Lipase@HOF-100 were close to zero, suggesting the inactivity of Lipase@HOF-100 towards p-NPP

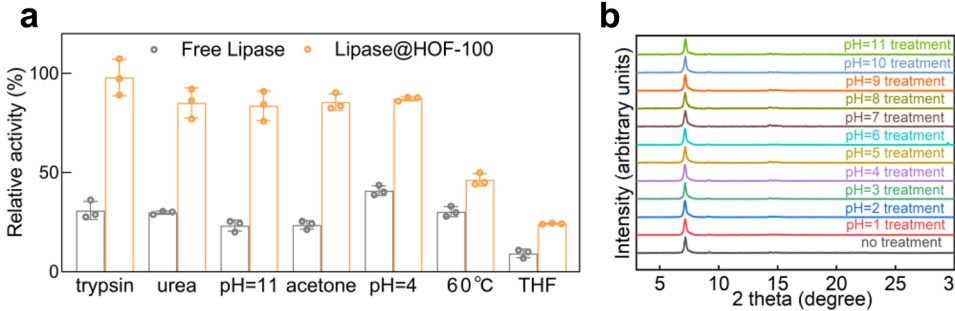

**Fig. 5 | Stability study of Lipase@HOF-100. a** The retained bioactivity of free Lipase and Lipase@HOF-100 after different treatments for 30 min. The catalytic substrate was NPB. The data of 3 independent experiments and calculated error bars (SD) are presented in **a**, SD standard deviation (*n* = 3). Data are presented as mean values +/−SD. **b** PXRD patterns of Lipase@HOF-100 after different pH treatment for 30 min.

(Fig. 4d and Supplementary Fig. 42, the absorbance decreases in the initial few seconds in Lipase@HOF-100 groups were caused by the transitory change in transmittance after adding the substrate solution into HOF particles). This hydrolysis selectivity of Lipase@HOF-100 was resulted from the rigorously gating effect of HOF-100 channel window, by which the large-sized p-NPP was excluded yet the small-sized NPB was permitted through. These results verified the gating concept of our HOF biocatalysts, which could provide important insight into the selective biocatalysis using HOF engineering.

Apart from the catalytic selectivity problem in some enzymes, the high susceptibility of natural enzyme also limits its diversity that can be explored in different research fields. Our highly crystalline HOF-100 scaffold not only imparted the enzymatic reaction high selectivity, but also could shield the fragile enzymes against non-physiological environments (Fig. 5a). Especially, after the treatment by proteolytic reagent (trypsin, 10 mg/mL), high concentrations of urea (6 M), basic water (pH=11), acetone and acidic water (pH=4) for 30 min, respectively, all Lipase@HOF-100 could retain more than 83% of original bioactivity using NPB as the catalytic substrate. However, the free Lipase suffered from serious deactivation after the similar treatments. Even though the bioactivities of Lipase@HOF-100 would decrease after exposure to high temperature (60 °C) and tetrahydrofuran (THF) solvent, it still surpassed the free Lipase after similar treatments.

The cryo-EM identified that the interlamellar spacing of HOF layer was ca. 0.34 nm (Supplementary Fig. 10), suggesting the strong layer-by-layer π-π stacking, which might afford the enzyme@HOF-100 biocatalyst high robustness. Indeed, the crystallinity (Fig. 5b) and morphology (Supplementary Fig. 43) of Lipase@HOF-100 were intact at a wide range pH from 1 to 11, indicating the highly structural stability. These results represented a prominent merit of HOF-100 biocatalysts for large-scale application in different pH scenarios.

**Ultrasensitive electrochemical immunosensor**

Combining with the biocatalytic function and structural stability, we constructed an ultrasensitive electrochemical immunosensor based on the cyclic biocascade mechanism, and used it for sensitive detection of mucin-1 (MUC1)[46,47], an important tumor biomarker (Fig. 6). The electrochemical immunosensing interface was built through the immobilization of capture antibody (Ab1) on the gold electrode (GE) via Au-NH₂ interaction. After blocking the non-specific sensing site by the saturation adsorption of BSA, the immunosensing electrodes were prepared (Fig. 6b). Then, the electrochemical beacon, termed as Ab2-Au-HRP@HOF-100 was engineered by anchoring gold nanoparticles (Au NPs) onto the pre-synthesized HRP@HOF-100 biocatalyst, wherein the anchored Au NPs promoted the combination with the second recognition agent (Ab2) onto HRP@HOF-100 biocatalyst through Au-NH₂ interaction (Fig. 6a). During the electrochemical beacon preparation, the zeta potentials of Au NPs and HRP@HOF-100 were measured to be −9.59 mV and −4.92 mV, respectively (Supplementary Fig. 44). It

suggested that Au NPs were hard to be combined onto HRP@HOF-100 owing to their electrostatic repulsion. To solve this problem, we employed a strong cationic polyelectrolyte, poly-(diallyldimethylammonium chloride) (PDDA), to modulate HRP@HOF-100 surface, resulting in a positively charged HRP@HOF-100-PDDA (20.80 mV). After that, the negatively charged Au NPs were easily to be anchored onto the positively charged HRP@HOF-100-PDDA through electrostatic interaction, along with the zeta potential change from 20.80 to 5.98 mV. These results indicated that the Ab2-Au-HRP@HOF-100 electrochemical beacon was successfully constructed, which was also supported by the HAADF-STEM imaging (Supplementary Fig. 45).

In the workflow of this electrochemical immunosensor, the electrodes were incubated with MUC1 samples and selectively captured MUC1 via an antibody−antigen interaction. Then, the as-prepared Ab2-Au-HRP@HOF-100 beacon was further anchored onto MUC1. After soaking this decorated electrode into electrolyte, the HRP@HOF-100 could be oxidized into HRP$_{Ox}$@HOF-100 under the action of H₂O₂, in which the generated HRP$_{Ox}$@HOF-100 was further catalyze the conversion of hydroquinone (HQ) to benzoquinone (BQ) and thus allow the regeneration of HRP@HOF-100 (Fig. 6b). Consequently, the produced BQ around electrode surface was re-reduced to produce the brilliant electrochemical emission. Through repeating the cyclic biocascade of H₂O₂-HRP@HOF-100-HQ, the magnified current signal was achieved and thus realized the ultrasensitive analysis of MUC1.

The successful development of this electrochemical immunosensor was demonstrated by the cyclic voltammogram (CV), electrochemical impedance spectroscopy (EIS) as well as differential pulse voltammetry (DPV) characterizations in Supplementary Fig. 46. Under the optimal experimental conditions (Supplementary Fig. 47), DPV responses toward various MUC1 concentrations were recorded to assess the sensitivity of electrochemical sensor. As demonstrated in Fig. 6c, the peak currents increased evenly with the enhancement of MUC1 concentrations from 1 pg/mL to 100 ng/mL. Based on the relationship between current signals as well as MUC1 concentrations, the linear regression equation (Fig. 6d) was described as $I = -7.107 \times \lg c - 8.418$, with the correlation coefficient ($R^2$) of 0.9996. According to the 3δ method[48], the value of the limit of detection (LOD) was calculated to be 0.18 pg/mL (detail seen in Supplementary Methods). Additionally, the analytical performances between this work and the previously reported methods for MUC1 determination was summarized in Supplementary Table 5. It could be seen that this proposed electrochemical immunosensor significantly outperformed the reported methods in terms of linear range and LOD, which was mainly ascribed to the highly biocatalytic activity and structural stability of the integrated HRP@HOF-100.

The insight into the specificity of this electrochemical immunosensing platform was examined by using MB, GOx and cholesterol as the interference models. Seen in Fig. 6e, the current signals of high concentrations of MB (100 ng/mL), GOx (100 ng/mL) and cholesterol (100 ng/mL) were equivalent to that of the blank group without MUC1,

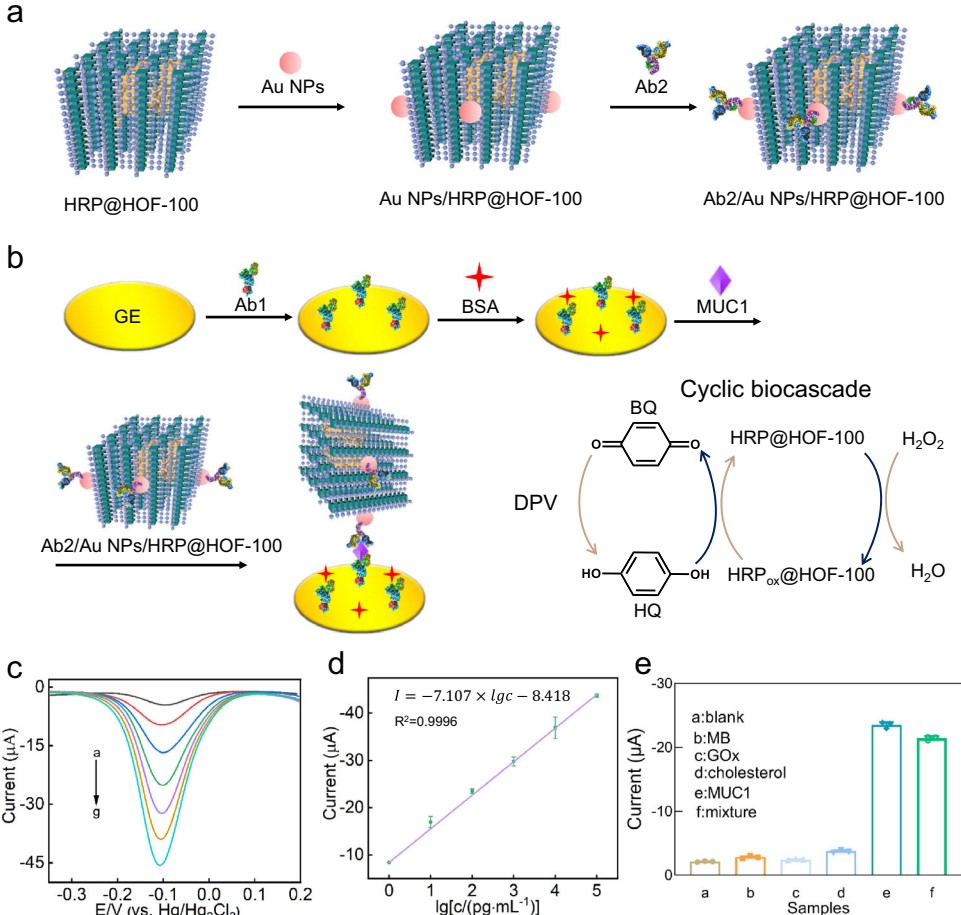

**Fig. 6 | Ultrasensitive electrochemical immunosensor. a** Schematic preparation of Ab2/Au NPs/HRP@HOF−100 electrochemical beacon. The pyrene core, formic acid arm and HRP in HRP@HOF-100 structure are shown as dark green prismatic square, purple ball and yellow surface model, respectively. **b** The principle of the electrochemical immunosensor based on cyclic biocascade for MUC1 detection. **c** DPV response profile of the developed electrochemical immunosensor as a function of MUC1: 0 (curve a), 1 pg/mL (curve b), 10 pg/mL (curve c), 100 pg/mL (curve d), 1 ng/mL (curve e), 10 ng/mL (curve f) and 100 ng/mL (curve g). **d** The obtained linear regression equation between the currents and the logarithm of different MUC1 concentrations. **e** Selectivity of the electrochemical immunosensor toward diverse samples: blank, MB (100 ng/mL), GOx (100 ng/mL), cholesterol (100 ng/mL), MUC1 (100 pg/mL) and a mixture solution (100 pg/mL MUC1, 100 ng/mL MB, 100 ng/mL GOx and 100 ng/mL cholesterol). The data of three independent experiments and calculated error bars (SD) are presented in **d**, **e**, SD standard deviation (*n* = 3). Data are presented as mean values +/−SD.

but much lower than the peak current of MUC1 solution (100 pg/mL) with 1000-fold low concentration as these interferences. It demonstrated the high specificity of our electrochemical immunosensor for MUC1 detection. Impressively, the mixture solution containing MUC1 (100 pg/mL) and other interferences with 1000-fold high concentration (100 ng/mL), presented a similar current signal as that of 100 pg/mL of MUC1 solution, indicating the ultrahigh anti-interference ability of our electrochemical immunosensor. Furthermore, the reproducibility was investigated by five parallel assays. Supplementary Fig. 48 displayed that all peak currents were comparable to each other, with a tiny relative standard deviation (RSD) value of 0.9%, manifesting a great reproducibility of this MUC1 electrochemical biosensor.

To further investigate the practicability of this HOF immunosensor, the standard addition method was performed to analyze MUC1 in four of 50-fold diluted health human blood serum samples. As displayed in Supplementary Table 6, the recoveries were in the range of 94%-107.6% and the corresponding RSD values were lower than 5.81%, which confirmed that the developed HOF immunosensing platform held great potentials for quantifying MUC1 in clinical diagnosis.

## Discussion
In conclusion, we reported a mild and green supramolecular strategy for bottom-up fabrication of hybrid biocatalyst. This spatial integration made full use of the catalytic activity of native enzyme and the confinement effect of highly crystalline nanoscaffold, and the practicability of this green method for designing efficient and stable biocatalysts was demonstrated in case of different enzymes. Impressively, the coplanar pyrene core of HOF building tectons offered a strong layer-by-layer, π-π-stacking interaction to stabilize the hybrid biocatalyst in a wide range of pH. This structurally explicit HOF scaffold with order-arranged pore channels could sever as the gating to sieve the catalytic substrate and thus enhance the selectivity of biocatalysis, and also could improve the resistance of enzymes in non-physiological environment. On basis of these attractive traits, an ultrasensitive electrochemical immunosensor was constructed based on the cyclic biocascade mechanism, which enabled the pg/mL detection of cancer biomarker. This work conquered the common stability and selectivity problems of biocatalysis using native enzymes, and offered insight to access the efficient hybrid biocatalysts for different applications.

## Methods
### Green synthesis of HOF hybrid biocatalysts
HOF hybrid biocatalysts were synthesized based on a green bottom-up strategy. 10 mg $H_4TCPy$ was dispersed in 0.3 mL of DMF under heating at 120 °C to get a clear solution. After cooling down to room temperature, 9 mL of aqueous solution containing 5 mg enzyme was

introduced. The mixed solution was stirred at room temperature for 5 min, followed by standing for another 15 min. The obtained HOF hybrid biocatalysts were collected by centrifugation at 13,680 × g, washed by deionized water two times and ethanol one time.

### Hydrolysis activity of Lipase using NPB/p-NPP as substrates

The hydrolysis activity assay of free Lipase/Lipase@HOF-100 was performed by the spectrometric method using p-nitrophenyl butyrate (NPB) and *p*-nitrophenyl palmitate (p-NPP) as substrates, respectively[6,49]. At the beginning, different masses of NPB and p-NPP were dissolved into pH=7.4 Tris-HCl/ethanol (1:3) solution to form different concentrations of substrate solutions. Then, 400 μL of free Lipase or Lipase@HOF-100 solution was added into a series concentration of 100 μL of substrate solutions. The Lipase content in each trail was kept at 100 μg/mL. The NPB and p-NPP could be hydrolyzed into the yellow product of p-nitrophenol (p-NP) by the catalyst of free Lipase or Lipase@HOF-100, and the generated p-NP could be traced at 405 nm by a UV-Vis spectrophotometer and quantified by a standard calibration curve (Supplementary Fig. 40).

### Fabrication of the electrochemical immunosensor

The bare gold electrode (GE, Φ = 3 mm) was pre-processed by alumina slurries (0.3 and 0.05 μm), and ultrasonicated in deionized water to remove the physical adsorbent[50]. Then, 5 μL of Ab1 (10 μg/mL) was deposited on the GE surface and incubated at 4 °C overnight. The Ab1-modified GE was further incubated with 5 μL of BSA (1%) solution for 60 min to block the nonspecific sites. After that, 10 μL of various concentrations of MUC1 solution was introduced to the above-modified electrode surface and incubated it for 120 min at 37 °C. Ultimately, 10 μL of well-dispersive aqueous solution of Ab2/Au NPs/HRP@HOF-100 biocomposite was placed onto the surface of the prepared electrode (MUC1/BSA/Ab1/GE), and incubated it for 90 min at 37 °C. Notably, the electrode was thoroughly rinsed with 0.1 M PBS solution (pH 7.4) after every modification step to remove weakly bound components.

### Electrochemical measurement procedure

For MUC1 determination, the DPV measurements were completed by a CHI 750E electrochemistry workstation in 0.1 M PBS (pH 7.4) embracing 4.5 mM HQ and 3.0 mM $H_2O_2$. Concretely, the DPV experimental parameters including initial potential, final potential, pulse height, step height and pulse width were 0.2 V, −0.35 V, 50 mV, 4 mV and 0.05 s, respectively. Both of CV and EIS measurements were carried out in 0.1 M KCl solution with 5 mM $[Fe(CN)_6]^{3−/4−}$. The corresponding potential range of CV was from -0.2 to 0.6 V with a scan rate of 50 mV/s, and the EIS measurement frequency ranged from 0.1 Hz to 100 kHz. All the electrochemical experiments in this work were conducted at room temperature.

### Statistics and reproducibility

The statistical analysis was performed in Origin Pro (version 2021), GraphPad Prism software (version 5.0.1), Microsoft Excel (version 2016) and DigitalMicrograph (Gatan) software (version 3.23.1518.0). For electronic and optical microscopy data in the main text (Figs. 2c, d; 3h; and 4a) and Supplementary Information (Supplementary Figs. 4a, b; 10a; 11a; 12a; 13a; 16a, b; 18; 20; 30; 34; 37; 43; and 45a, b), more than three repeats in each experiment were carried out independently with similar results.

### Reporting summary

Further information on research design is available in the Nature Portfolio Reporting Summary linked to this article.

## Data availability

All data supporting this study and its findings are available within the article and its Supplementary Information or from the corresponding authors upon request. The HRP structure used herein is available in the PDB database under accession code 1hch. The GOx structure used herein is available in the PDB database under accession code 1gal. The Lipase structure used herein is available in the PDB database under accession code 1trh.

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

## Acknowledgements

We acknowledge financial support from projects of the National Natural Science Foundation of China (22174164, G.C.; 22104159, Siming Huang), the Natural Science Foundation of Guangzhou City (202201011408, G.C.), and Fundamental Research Funds for the Central Universities, Sun Yat-sen University (23lgbj005, G.C.).

## Author contributions

G.C. conceived the idea, designed the experiments and wrote the manuscript. W.H. performed material synthesis, characterization and wrote the manuscript. H. Yuan, H. Yang, Shuyao Huang, and H.Z. helped with the material synthesis, characterization, and data analysis. X.M. performed the cryo-EM experiment and helped with the data analysis. G.C., Siming Huang, and G.O. supervised the experiments and provided financial support.

## Competing interests

The authors declare no competing interests.
