## [Peer Review File · Nature Communications]

Green synthesis of stable hybrid biocatalyst using a hydrogen-bonded, π - π -stacking supramolecular assembly for electrochemical immunosensorREVIEWER COMMENTS

Reviewer #1 (Remarks to the Author):

In this paper, Ouyang, Chen, and co-workers designed a hybrid biocatalyst with the hydrogen-bonded organic framework (HOF-100) as an enzyme encapsulation carrier, which showed superior biocatalytic function and structural stability. Subsequently, an electrochemical immunosensor capable of detecting pg/mL cancer biomarkers was constructed. This paper may broaden the application of hybrid biocatalysts in the field of biosensing and immunoassay. There are some issues need to be concerned.

1. The authors indicated that when 9 mL of HRP aqueous solution (0.54 mg/mL) was added into 300 μ L of well-dispersive H4TCPy dimethyl formamide (DMF) solution, a mass of biocomposites were generated as shown in Figure S1. What is the phenomenon without enzyme solution? The authors are suggested to provide additional experiments of enzyme-free assembly process for control.
2. The authors carried out an enzyme-adsorption experiment to elucidate that the bulky HRP was unable to infiltrate into the micropores of HOF-100 or surface-adsorb onto HOF-100. However, there are significant differences in the PXRD patterns of HOF-100 before and after HRP adsorption in Figure S7c. While, the PXRD pattern of HOF-100 absorbed HRP was similar to that of HRP@HOF-100. These results were unable to demonstrate the HRP was fused into HOF-100. Besides, there were no differences in PXRD patterns of HOF-100 before and after lipase adsorption in Figure S27c. The authors should explain that phenomenon.
3. The authors have stated the red fluorescence overlapped perfectly with the HOF-100 nano scaffold confirmed the fused HRP was uniformly confined within the framework in Figure 1d. However, CLSM images could not show the rod-like nanostructure of HRP@HOF-100. A similar report from Christian J. Doonan and co-workers (J. Am. Chem. Soc. 2019, 141, 36, 14298-14305) has shown much clear CLSM images. Therefore, the authors should revise those images.
4. The authors stated that the typical I-type curves revealed the microporous structure of the HRP@HOF-100. It should be better to deliver the pore size distribution data to further demonstrate pore structure.
5. The authors stated that, under the [10-1] project, the rhombus-shaped channel windows with 1.2 \times 0.8 nm dimension were clearly captured under a low-electron-dose imaging model (Figure 3a), which were in line with the theoretical model of HOF-100. What is the state of lipase in this case? Does it hinder the self-assembly of H4TCPy tectons? What is the effect of lipase on the channel window and crystallinity of HOF-100? The authors are suggested to clarify more.
6. In Figures 3d and S33, the catalytic kinetics curves of lipase@HOF-100 displayed significant decreases in the initial few seconds. More discussion should be given.
7. The authors should point out the type of catalytic substrate in Figure 4a.
8. The authors stated the loading of Au NPs was verified by the zeta potentials test in Figure S36. The authors should further explain this result and give the full name of PDDA.
9. In Figure 5b, the name of the gold electrode on the figure does not match the description. The authors are suggested to correct it.
10. The authors have published several papers on enzyme@HOFs. The significance of the paper are

suggested to introduce more.

Reviewer #2 (Remarks to the Author):

The concept of integration of an enzyme and porous molecular crystal was first proposed by Sumbly White, Falcaro, and Doonan in 2019, as the authors cited in Ref.27, and further related papers has been published such as refs.28-30. From this sense, novelty of the present manuscript seems to be lower than the previous reports. On the other hand, the authors thoroughly conducted characterization of the hybrid materials, extended this concept to seven kinds of proteins, and showed important proof-of-concepts for selective and condition tolerant catalysts and an electrochemical immunosensor for sensitive detection of mucin-1 as follows. First, they demonstrated through structural characterization of the hybrid material of Horseradish peroxidase HRP. The hybrid material was prepared just by mixing aqueous solution of the enzyme and DMF solution of the building block molecule H4TCPy, which is an environmentally friendly process. Fusion of the protein was confirmed by IR absorption of the amide I band of the protein. CLSM and ED also supported formation of the fused HOF. Porosity of the HOF part was evaluated by gas sorption experiments to show type-I sorption isotherm with BET surface area of 571 m²/g. The resultant material is stable in a range of pH from 1 to 11. The confirmation of the enzyme was confirmed by CD analysis. These thorough experiments well support the authors conclusion.

Next, they also demonstrated size selective and condition-tolerant catalytic reactions using lipase@HOF-100. Furthermore, the authors incorporated the HRP-HOF-100 into an electrochemical immunosensor for sensitive detection of mucin-1.

These results strongly indicate powerful potential of HOF hybrid systems for applicable materials and should be very impressive for broad range of readers. Therefore, the manuscript can be suitable for publication in Nature Communications after the following comments are considered.

1. The reviewer wants to know if the enzyme makes an aggregate or is isolated one by one in the HOF lattice. Have the authors tried to reveal a distribution manner of the enzyme in the framework at the molecular level.

2. In Figure 3c, the chemical structure of NPB may be wrong.

Reviewer #3 (Remarks to the Author):

In this manuscript, Chen and Ouyang et al report a supramolecular HOF strategy to design crystalline hybrid biocatalysts and explore their uses in size-sieving biocatalysis and immunosensor. By virtue of designing pyrene-based molecular tecton, the proteins with different surface chemistries are readily fused into a highly crystalline crystal under a green and biocompatible process. This process is almost solvent-free and time-saving. The protein conformations and their biocatalytic activities are demonstrated to be well maintained thanks to this mild crystallization. While the porous crystal shell imparts the fused enzyme (lipase) with higher substrate selectivity and stability for ester hydrolysis, attributing to the nano-confinement effect of structurally unambiguous HOF.

I am impressive that this supramolecular strategy is workable to different enzymes and the formed hybrid biocatalysts are structurally stable in a wide range of pH owing to the pi-pi stacking of pyrene core. This leads to some exciting functions in selective biocatalysis and immunosensor applications. Even though the concept of enzyme immobilization using HOF has been reported as the author noted, this work provides significant advances in terms of enzyme loading, structural stability and biocatalytic activity. It is also interesting for me to see the possibility of HOF crystals in selective biocatalysis and electrochemical immunosensor area, which has been dominated by MOF/COF for long time.

Overall, this work is well organized and the conclusion is supported by the experimental data. I recommend the acceptance of this manuscript in *Nat. Commun.* after suitable revisions.

- 1、 The HOF structure is imaged by the cryo-EM. This is an important method to identify the microstructure of this hybrid biocatalyst. The cryo-EM parameters (electron dosage rate, exposure time, etc.) should be provided in the main text.
- 2、 It is very impressive to see the efficiency and practicability of this HOF crystal for different proteins. But I wonder why the loading of lipase is calculated based on the standard BCA assay, and the loading of other proteins are calculated based on the standard Bradford assay.
- 3、 It is necessary to employ the CD spectra to further prove the conformations of GOx and Cyt c do not change after incubating in DMF aqueous solution for 6h.
- 4、 The application of HOF crystals in electrochemical immunosensor area is new. I suggest to revise the title to highlight this point.
- 5、 The authors only mention that the method for calculating the detection limit is based on the 3 σ principle, but the specific calculation method should also be clearly stated.
- 6、 All figures both in the main text and in Supporting Information should be described and discussed, such as Figure S35 and Figure S36.
- 7、 The captions of Figure 1a/Figure S21 should be changed to“PXRD patterns of HRP@HOF-100/lipase@HOF-100, experimental HOF-100 and simulated HOF-100.”.
- 8、 Abbreviations should be clearly defined when they first appear, including but not limited to: SEM.
- 9、 The GC in Figure 4b should be modified to GE, which is in accordance with the text.

Response to the reviewers' comments

We thank the reviewers for their constructive and insightful comments, which we have addressed in detail in this point-by-point response. The reviewers' comments are in **black** with authors' point-by-point responses in **blue**. Changes to the manuscript are highlighted in **yellow** in the revised main text and Supplementary Information files.

Reviewer #1 (Remarks to the Author):

In this paper, Ouyang, Chen, and co-workers designed a hybrid biocatalyst with the hydrogen-bonded organic framework (HOF-100) as an enzyme encapsulation carrier, which showed superior biocatalytic function and structural stability. Subsequently, an electrochemical immunosensor capable of detecting pg/mL cancer biomarkers was constructed. This paper may broaden the application of hybrid biocatalysts in the field of biosensing and immunoassay. There are some issues need to be concerned.

Response: We thank for the reviewer's positive comments, and we have endeavored to address the issues according to your insightful comments.

1. The authors indicated that when 9 mL of HRP aqueous solution (0.54 mg/mL) was added into 300 μ L of well-dispersive H₄TCPy dimethyl formamide (DMF) solution, a mass of biocomposites were generated as shown in Figure S1. What is the phenomenon without enzyme solution? The authors are suggested to provide additional experiments of enzyme-free assembly process for control.

Response: We thank for the reviewer's kindly suggestion. We have conducted the control experimental by adding equivalent water instead of enzyme aqueous solution (**Supplementary Fig. 2**). As like the enzyme assembly process (**Supplementary Fig. 1**), the precipitates could be formed in this enzyme-free assembly process. This indicated that the enzyme was encapsulated through a co-precipitation process (*Nano Lett.* 2014, 14, 5761–5765).

We have provided the control experiment in **Supplementary Fig. 2**.

Supplementary Fig. 1. Recording the assembly process by photograph. (a) when HRP was introduced into the H₄TCPy solution; (b) after 5 min reaction.

Supplementary Fig. 2. Recording the assembly process by photograph. (a) when H₂O was introduced into the H₄TCPy solution; (b) after 5 min reaction.

2. The authors carried out an enzyme-adsorption experiment to elucidate that the bulky HRP was unable to infiltrate into the micropores of HOF-100 or surface-adsorb onto HOF-100. However, there are significant differences in the PXRD patterns of HOF-100 before and after HRP adsorption in Figure S7c. While, the PXRD pattern of HOF-100 absorbed HRP was similar to that of HRP@HOF-100. These results were unable to demonstrate the HRP was fused into HOF-100. Besides, there were no differences in PXRD patterns of HOF-100 before and after Lipase adsorption in Figure S27c. The authors should explain that phenomenon.

Response: We are sorry for this mistake. we have re-conducted the adsorption experiment (**Supplementary Figs. 9 and 36**). In fact, the main diffraction peaks at $2\theta=7$ were recorded in raw HOF-100, HRP-adsorbed HOF and HRP@HOF, and this was similar as the Lipase adsorption. These suggested that the adsorption process could not affect the crystallographic structure of HOF host. Meanwhile, the stability experiment confirmed that our HOF-100 biocatalyst could be stable in a wide pH ranged from 1 to 11, also indicated the high stability of HOF (**Fig. 4b**). We have provided the new PXRD data in **Supplementary Figs. 9 and 36** in the revised Supplementary Information and given more discussion in the Figure Caption.

Supplementary Fig. 9c. The PXRD patterns of HOF-100 before and after HRP adsorption.

Supplementary Fig. 36c. The PXRD patterns of HOF-100 before and after Lipase adsorption.

In addition, we also carried out several experiments to confirm the fusion of enzyme. In Fourier-transform infrared (FTIR) spectra (**Supplementary Fig. 6**), the characteristic peak at $1700\text{--}1610\text{ cm}^{-1}$, regarded as the typical amide I band of an enzyme, was observed in HRP@HOF-100, but not in standard HOF-100. This given the first evidence of the fusion of HRP. In addition, the fusion of HRP was also endorsed by the thermogravimetric analysis (TGA), wherein the weight loss at ca. $150\text{--}350\text{ }^{\circ}\text{C}$ was caused by the pyrolysis of proteins (**Supplementary Fig. 7**). The insight into the porous structure was further examined by nitrogen adsorption/desorption isotherm experiment (**Fig. 1b**). The typical I-type curves revealed the microporous structure of the as-synthesized HRP@HOF-100 (**Fig. 1b and Supplementary Fig. 8**), with a calculated Brunner-Emmet-Teller (BET) surface area as high as $571.2\text{ m}^2/\text{g}$. Nevertheless,

this BET value was decreased compared to the pure HOF-100 (780.1 m²/g), manifesting that the interior pore of HOF-100 was partially occupied by HRP.

To visualize the spatial distribution of HRP, we re-performed the confocal laser scanning microscope (CLSM) experiment, in which HRP was pre-labelled with the red dye, rhodamine B (RhB). As demonstrated in **Fig. 1d**, the red fluorescence overlapped perfectly with the HOF-100 nanoscaffold, confirming the fused HRP was uniformly confined within the framework. Consequently, all the aforementioned results verified that this new in situ HOF approach permitted the high-efficient enzyme encapsulation under biocompatible environment.

Fig. 1d. The CLSM images showing the spatial distribution of HRP (HRP was labelled by red dye) within HOF-100 scaffold.

3. The authors have stated the red fluorescence overlapped perfectly with the HOF-100 nano scaffold confirmed the fused HRP was uniformly confined within the framework in Figure 1d. However, CLSM images could not show the rod-like nanostructure of HRP@HOF-100. A similar report from Christian J. Doonan and co-workers (J. Am. Chem. Soc. 2019, 141, 36, 14298-14305) has shown much clear CLSM images. Therefore, the authors should revise those images.

Response: Thank you for the reviewer's useful comment. Comparison with the rod-like HOF reported by Christian J. Doonan and co-workers (J. Am. Chem. Soc. 2019, 141, 36, 14298-14305), our HOF biocatalyst has much smaller dimension. Owing to the limited resolution of CLSM, the rod-like structure of our HOF biocatalyst was not so clear.

According to the reviewer's suggestion, we have repeated the CLSM experiment and provided the high-resolution CLSM images in **Fig. 1d**, which could show the rod-like nanostructure of HRP@HOF-100.

We have provided the new CLSM image in **Fig. 1d** in revised manuscript.

Fig. 1. Preparation and structural characterization of HOF hybrid biocatalysts. (a) PXRD patterns of HRP@HOF-100, experimental HOF-100 and simulated HOF-100. (b) Nitrogen adsorption/desorption isotherm of HRP@HOF-100 and pure HOF-100. (c) The cryo-EM image showing the long-range ordered channels within HRP@HOF-100. The inset was the fast Fourier transform image. (d) The CLSM images showing the spatial distribution of HRP (HRP was labelled by red dye) within HOF-100 scaffold.

4. The authors stated that the typical I-type curves revealed the microporous structure of the HRP@HOF-100. It should be better to deliver the pore size distribution data to further demonstrate pore structure.

Response: Thank you for the reviewer's constructive comments. According to the reviewer's suggestion, we have calculated the pore-size distribution based on NLDFT method, and the result was displayed in **Supplementary Fig. 8**.

Both HOF-100 and HRP@HOF-100 showed the micropores centered at 0.73 nm and 1.2 nm, which were in line with the theoretical pore dimension (0.8 nm × 1.2 nm). Compared to HOF-100, HRP@HOF-100 showed a decreased pore volume. This was caused by the spatial occupation by the enzymes, which further demonstrated that the enzymes were indeed encapsulated into, rather than surface-adsorbed onto the HOF-100.

We have provided the pore size distribution (**Supplementary Fig. 8**) and relative discussion in the revised manuscript.

Supplementary Fig. 8. The pore-size distributions of HOF-100 and HRP@HOF-100 based on NLDFT.

5. The authors stated that, under the [10-1] project, the rhombus-shaped channel windows with 1.2×0.8 nm dimension were clearly captured under a low-electron-dose imaging model (Figure 3a), which were in line with the theoretical model of HOF-100. What is the state of Lipase in this case? Does it hinder the self-assembly of H_4 TCPy tectons? What is the effect of Lipase on the channel window and crystallinity of HOF-100? The authors are suggested to clarify more.

Response: Thank you for this insightful suggestion. We adopted the cryo-EM to identify the microstructure of HOF-100 after Lipase encapsulation. It showed that the obtained Lipase@HOF-100 have long-range ordered microchannel (ca. 1.2 nm width) throughout the biocatalyst (**Supplementary Fig. 37**). In addition, under the [10-1] project, the rhombus-shaped channel windows with 1.2×0.8 nm dimension were clearly captured under a low-electron-dose imaging model (**Fig. 3a**), which was well agreement with the theoretical HOF-100 structure. This gave the first evidence that the Lipase could not affect the assembly of H_4 TCPy tectons and the channel window and crystallinity of HOF-100.

Supplementary Fig. 37. cryo-EM image of Lipase@HOF-100. The cryo-EM image showing the long-range ordered channels within Lipase@HOF-100.

In addition, the PXRD revealed that both HOF-100 and Lipase@HOF-100 have identical PXRD patterns

with sharp characteristic peaks (**Supplementary Fig. 36c**), further supporting the fact that the Lipase could not affect the crystallinity of HOF-100. Furthermore, both the N₂ isotherms data and the calculated pore size distribution also demonstrated that the high porosity was retained after Lipase encapsulation, although the pore volume was decreased attributing to the spatial occupation of Lipase (**Supplementary Fig. 31**).

Supplementary Fig. 31. The nitrogen adsorption/desorption isotherms and pore-size distributions of HOF-100 and Lipase@HOF-100. (a) Nitrogen adsorption/desorption isotherm of Lipase@HOF-100 and pure HOF-100. (b) The pore-size distributions of HOF-100 and Lipase@HOF-100 based on NLDFT method.

It is still challenging to probe an enzyme's confirmation in a solid carrier. According to the reviewer's suggestion, we attempted to understand the state of Lipase in HOF biocatalyst by different spectroscopic methods. The FT-IR (**Supplementary Fig. 32**, TGA (**Supplementary Fig. 33**) and CLSM experiments (**Supplementary Fig. 34**) confirmed the successful incorporation of Lipase. In addition, the CD spectrum showed that the conformation of Lipase could be well preserved in this mild assembling condition (**Supplementary Fig. 38**). The insight into the biointerface was examined by ssNMR (**Supplementary Fig. 39**). The chemical shift at 12.7 ppm in H1 ssNMR of Lipase@HOF-100 biocatalyst is assigned to the proton of the carboxyl groups of H₄TCPy molecular tectons. This peak is observed to be shifted into a low magnetic field compared with the physical mixed sample of Lipase and HOF-100 ($\delta=11.5$ ppm), suggesting the biointerface interaction between Lipase and the carboxyl groups of HOF-100 scaffold. We believed that such biointerface should play an important role on the high encapsulation efficiency during the bottom-up assembly.

Supplementary Fig. 38. The CD spectra. The CD spectra of free Lipase in water and Lipase after

incubating in DMF aqueous solution for 6 h. The volumes of DMF and deionized water were 0.3 mL and 9.0 mL, respectively, which was in agreement with the assembly system.

Supplementary Fig. 39. The ssNMR spectra of HOF-100 and Lipase@HOF-100. The ssNMR spectra of Lipase@HOF-100 biocomposite and the physically mixed sample of Lipase and HOF-100.

Even though we could not give more information on the state of Lipase at current stage, we have confirmed that in the case of small-sized ester (NPB), the Lipase@HOF-100 biocatalyst showed comparable catalytic ability to free Lipase (**Fig. 3c**). This indicated that the Lipase tended to maintain its native catalytic function after assembling.

Fig. 3c. Plot of reaction velocity, V , against substrate [NPB]. The dosages of for Lipase and Lipase@HOF-100 were kept at the same in each trial (80 $\mu\text{g}/\text{mL}$).

We have provided the relative discussion in the main text and supporting information.

6. In Figures 3d and S33, the catalytic kinetics curves of Lipase@HOF-100 displayed significant decreases in the initial few seconds. More discussion should be given.

Response: Many thanks for this constructive comment. The decreases in the initial few seconds of Lipase@HOF-100 group was caused by the transitory change in transmittance after adding the substrate solution into HOF particles. We have provided the interpretation in the revised manuscript (**Fig. 3d and Supplementary Figs. 41 and 42**).

7. The authors should point out the type of catalytic substrate in Figure 4a.

Response: We greatly appreciate the reviewer's kind reminder. The catalytic substrate is NPB, and we have clarified it in the Figure Caption of **Fig. 4a** in the revised manuscript.

8. The authors stated the loading of Au NPs was verified by the zeta potentials test in Figure S36. The

authors should further explain this result and give the full name of PDDA.

Response: Thanks a lot for the reviewer's valuable comments. We have provided the detail discussion of the zeta potentials in **Supplementary Fig. 44**. The zeta potentials of Au NPs and HRP@HOF-100 were measured to be -9.59 mV and -4.92 mV, respectively. It suggested that Au NPs were hard to be combined with HRP@HOF-100 owing to their electrostatic repulsion. To solve this problem, we employed a strong cationic polyelectrolyte, poly-(diallyldimethylammonium chloride) (PDDA), to modulate HRP@HOF-100, and it formed a positively charged HRP@HOF-100-PDDA (20.80 mV). After that, the negative Au NPs were easily to be anchored onto positive HRP@HOF-100-PDDA through electrostatic interaction, along with the zeta potential shifted from 20.80 mV to 5.98 mV. These results indicated the Au NPs were successfully combined with the HRP@HOF-100.

In addition, we have given the full name of PDDA as poly-(diallyldimethylammonium chloride) in the main text.

Supplementary Fig. 44. The Zeta potentials. The Zeta potentials of Au NPs, HRP@HOF-100, HRP@HOF-100-PDDA and Au NPs/HRP@HOF-100. Data are presented as mean \pm s.d. (error bars); $n = 3$.

9. In Figure 5b, the name of the gold electrode on the figure does not match the description. The authors are suggested to correct it.

Response: We are sorry for this mistake. We have corrected it in **Fig. 5b**.

Fig. 5. Ultrasensitive electrochemical immunosensor. (a) Schematic preparation of Ab2/Au NPs/HRP@HOF-100 electrochemical beacon. (b) The principle of the electrochemical immunosensor based on cyclic biocascade for MUC1 detection. (c) DPV response profile of the developed electrochemical immunosensor as a function of MUC1: 0 (curve a), 1 pg/mL (curve b), 10 pg/mL (curve c), 100 pg/mL (curve d), 1 ng/mL (curve e), 10 ng/mL (curve f) and 100 ng/mL (curve g). (d) The obtained linear regression equation between the currents and the logarithm of different MUC1 concentrations. Data are presented as mean \pm s.d. (error bars); $n = 3$. (e) Selectivity of the electrochemical immunosensor toward diverse samples: blank, MB (100 ng/mL), GOx (100 ng/mL), cholesterol (100 ng/mL), MUC1 (100 pg/mL) and a mixture solution (100 pg/mL MUC1, 100 ng/mL MB, 100 ng/mL GOx and 100 ng/mL cholesterol). Data are presented as mean \pm s.d. (error bars); $n = 3$

10. The authors have published several papers on enzyme@HOFs. The significance of the paper are suggested to introduce more.

Response: Many thanks for this comment. We are sorry for this unclear description. In the previous report by our group, it required milliliters of organic solvent to disperse the pyrene tectons and thus enabled the assembly of enzyme@HOF. This organic solvent dosage may limit the practicability of the reported method to some fragile enzymes. In this work, by means of modulating the carboxylic arm of pyrene core tecton, it allowed the one-pot assembly of enzyme@HOF-100 biocatalyst in an almost organic solvent-free approach (only 300 μL organic solvent was required), and this green synthesis protocol was applicable to different proteins including enzymes. In addition, the highly crystalline HOF-100 network showcased the first example of substrate-sieving biocatalysis. Last but not least, the HOF electrochemical immunosensor was reported for the first time based on the rational design of enzyme@HOF-100, and this

field has been dominated by MOF/COF immunosensor for long time (also mentioned by reviewer 3).

We have added more discussion to highlight the importance of our work in the Introduction Section.

Reviewer #2 (Remarks to the Author):

The concept of integration of an enzyme and porous molecular crystal was first proposed by Sumbly White, Falcaro, and Doonan in 2019, as the authors cited in Ref.27, and further related papers has been published such as refs.28-30. From this sense, novelty of the present manuscript seems to be lower than the previous reports. On the other hand, the authors thoroughly conducted characterization of the hybrid materials, extended this concept to seven kinds of proteins, and showed important proof-of-concepts for selective and condition tolerant catalysts and an electrochemical immunosensor for sensitive detection of mucin-1 as follows. First, they demonstrated through structural characterization of the hybrid material of Horseradish peroxidase HRP. The hybrid material was prepared just by mixing aqueous solution of the enzyme and DMF solution of the building block molecule H4TCPy, which is an environmentally friendly process. Fusion of the protein was confirmed by IR absorption of the amide I band of the protein. CLSM and ED also supported formation of the fused HOF. Porosity of the HOF part was evaluated by gas sorption experiments to show type-I sorption isotherm with BET surface area of 571 m²/g. The resultant material is stable in a range of pH from 1 to 11. The confirmation of the enzyme was confirmed by CD analysis. These thorough experiments well support the authors conclusion.

Next, they also demonstrated size selective and condition-tolerant catalytic reactions using Lipase@HOF-100. Furthermore, the authors incorporated the HRP-HOF-100 into an electrochemical immunosensor for sensitive detection of mucin-1.

These results strongly indicate powerful potential of HOF hybrid systems for applicable materials and should be very impressive for broad range of readers. Therefore, the manuscript can be suitable for publication in Nature Communications after the following comments are considered.

Response: We thank for the reviewer's positive comments, and we have endeavored to address the issues according to your insightful comments.

1. The reviewer wants to know if the enzyme makes an aggregate or is isolated one by one in the HOF lattice. Have the authors tried to reveal a distribution manner of the enzyme in the framework at the molecular level.

Response: Thanks a lot for this constructive comment. It is still challenging to probe the state, especially the dispersibility, of an enzyme confined in an HOF crystal, because of the chemical composition between proteins and HOF is quite similar. As far as we known, CLSM is a widely used method to study the spatial distribution of an enzyme in a porous crystal (Chem. Rev. 2021, 121, 1077-1129; Chem, 2016, 1, 154-169). Yet, limiting in the resolution of CLSM, this method is unable to identify the dispersibility of an enzyme.

Seeing is believe. The high-resolution TEM is a powerful technique to study the spatial location of guest within a solid carrier. However, owing to the low contrast of light element (C, N, O) of protein under TEM image, it is still impossible to probe the protein location using TEM so far.

In order to locate the protein under TEM, we attempted to label the enzyme with nanoclusters (NCs) of heavy element, wherein the Pt NCs was in situ reduced within an enzyme using a compatible method (Fig. 2g, Angew. Chem. Int. Ed. 2017, 56, 6767-6772). This method afforded the Pt NCs-labelled enzyme yet had limited influence on the conformation of an enzyme (Supplementary Fig. 15). As shown in Supplementary Fig. 16a, the as-synthesized Pt NCs with ca. 2.5 nm dimension was monodisperse, indicating the stabilization by enzyme. We further confirmed the successful labeling of Pt NCs by a protein

negative staining experiment (detail seen in Supplementary Method: 1.11). Under the protein negative staining experiment, we clearly observed the enzyme outline (in white) around individual Pt NCs, suggesting each enzyme was successfully labelled by a Pt NCs (**Supplementary Fig. 16b**). In addition, the insight into the molecular weight MALDI-TOF MS showed that the enzyme weight increased from 76794 to 78606 after labeling (**Supplementary Fig. 17**), and it gave a labelling efficiency of ca. nine Pt per one enzyme. These results adequately demonstrated the highly efficient labelling of Pt NCs in enzyme.

Fig. 2g. Schematic representation of the Pt NCs labelling process.

GOx		GOx-Pt NCs			
	Fraction	Ratio	Fraction	Ratio	
Helix	0.0	22.2	Helix	0.0	16.6
Beta	0.1	46.0	Beta	0.1	54.4
Turn	0.0	4.9	Turn	0.0	3.9
Random	0.0	26.8	Random	0.0	25.1
Total	0.2	100.0	Total	0.2	100.0
RMS Value	10.956		RMS Value	9.795	

Supplementary Fig. 15. The CD spectra. The CD spectra of free GOx and GOx-Pt NCs (left), and the calculated secondary structure contents based on Yang's reference (right).

Supplementary Fig. 16. The TEM images of GOx-Pt NCs before and after negative staining. (a) The TEM image of GOx-Pt NCs. (b) The TEM image of GOx-Pt NCs after the protein negative staining treatment.

Supplementary Fig. 17. The MALDI-TOF analysis. MALDI-TOF analysis of the molecular weight of GOx and GOx-Pt NCs.

Under the cryo-EM in Low-magnification, we got a glimpse of the spatial location of enzyme within HOF-100. The Pt-labelled enzyme (GOx-Pt nanoclusters, GOx-Pt NCs) was discretely located in HOF-100

scaffold (**Supplementary Fig. 18**). In addition, in the high-magnification cryo-EM imaging, we identified the highly crystalline HOF-100 pores and many monodispersed Pt NCs (highlighted in red circles) simultaneously (**Fig. 2h**).

Supplementary Fig. 18. The cryo-EM image of GOx-Pt@HOF-100. Low-magnification cryo-EM image of GOx-Pt NCs -encapsulated HOF-100.

Fig. 2h. High-magnification cryo-EM image of GOx-Pt NCs-encapsulated HOF-100.

Based on this labelling experiment, we inferred that the encapsulated enzyme was discretely located in HOF-100. And both the high dispersibility of enzyme and the opening pore channel of HOF-100 favor the biocatalysis in a confined microenvironment.

We have added these data in the revised manuscript.

2. In Figure 3c, the chemical structure of NPB may be wrong.

Response: Thanks a lot for the reviewer's careful reading. We have revised the incorrect chemical structure of NPB in **Fig. 3c**.

Fig. 3. HOF Gating effect for enhancing the biocatalytic selectivity. (a) The cryo-EM image identifying the rhombus-shaped pore windows from the [10-1] projection, and the inset is the enlarged domain. (b) The schematic illustration of the gating effect of Lipase@HOF-100 for ester hydrolysis. (c) Plot of reaction velocity, V , against substrate [NPB]. The dosages of for Lipase and Lipase@HOF-100 were kept at the same in each trial (80 $\mu\text{g}/\text{mL}$). (d) The catalytic kinetics curves of free Lipase and Lipase@HOF-100 under 0.2 mM $p\text{-NPP}$. The dosages of Lipase (80 $\mu\text{g}/\text{mL}$) used were kept at the same in each trial.

Reviewer #3 (Remarks to the Author):

In this manuscript, Chen and Ouyang et al report a supramolecular HOF strategy to design crystalline hybrid biocatalysts and explore their uses in size-sieving biocatalysis and immunosensor. By virtue of designing pyrene-based molecular tecton, the proteins with different surface chemistries are readily fused into a highly crystalline crystal under a green and biocompatible process. This process is almost solvent-free and time-saving. The protein conformations and their biocatalytic activities are demonstrated to be

well maintained thanks to this mild crystallization. While the porous crystal shell imparts the fused enzyme (Lipase) with higher substrate selectivity and stability for ester hydrolysis, attributing to the nano-confinement effect of structurally unambiguous HOF.

I am impressed that this supramolecular strategy is workable to different enzymes and the formed hybrid biocatalysts are structurally stable in a wide range of pH owing to the π - π stacking of pyrene core. This leads to some exciting functions in selective biocatalysis and immunosensor applications. Even though the concept of enzyme immobilization using HOF has been reported as the author noted, this work provides significant advances in terms of enzyme loading, structural stability and biocatalytic activity. It is also interesting for me to see the possibility of HOF crystals in selective biocatalysis and electrochemical immunosensor area, which has been dominated by MOF/COF for long time.

Overall, this work is well organized and the conclusion is supported by the experimental data. I recommend the acceptance of this manuscript in Nat. Commun. after suitable revisions.

Response: We thank for the reviewer's positive comments, and we have endeavored to address the issues according to your insightful comments.

1. The HOF structure is imaged by the cryo-EM. This is an important method to identify the microstructure of this hybrid biocatalyst. The cryo-EM parameters (electron dosage rate, exposure time, etc.) should be provided in the main text.

Response: Thanks a lot for the reviewer's kind reminder. The optimal parameters were described as follow: The dose rate was ~ 15 counts/pixel/second, and the exposure time in a frame was 0.023 s. Each micrograph stack contains 10 frames (the total exposure time was 0.23 s), and the total dose rate was ca. $30 \text{ e}^-/\text{\AA}^2$ per micrograph. We have added the cryo-EM parameters in main text and supplementary method section in the revised manuscript.

2. It is very impressive to see the efficiency and practicability of this HOF crystal for different proteins. But I wonder why the loading of Lipase is calculated based on the standard BCA assay, and the loading of other proteins are calculated based on the standard Bradford assay.

Response: Thanks for the reviewer's valuable comment. Both bicinchoninic acid (BCA) and Bradford assays are the extensively methods for protein quantification. In fact, we tried to quantify the Lipase using standard Bradford assay, but we found that the standard Bradford assay was not workable for Lipase. On the contrary, using standard BCA assay, the concentration of Lipase was in direct proportion to the UV-vis absorbance at 562 nm. Therefore, we chose the standard BCA assay for Lipase quantification.

We have added this description in the Supplementary method: **1.7**.

3. It is necessary to employ the CD spectra to further prove the conformations of GOx and Cyt c do not change after incubating in DMF aqueous solution for 6h.

Response: We highly appreciate the reviewer's insightful suggestion. The CD spectra of GOx and Cyt c after incubating in DMF (300 μL) aqueous solution for 6 h have been provided in **Supplementary Figs. 24 and 26**. Both the CD profiles of Cyt c and GOx were well retained, and the calculated secondary structure by Yang's reference also confirmed the negligible effect on the enzyme conformation by this treatment.

Cyt c in water			Cyt c in DMF/water		
	Fraction	Ratio		Fraction	Ratio
Helix	0.0	33.9	Helix	0.0	34.0
Beta	0.0	10.6	Beta	0.0	7.0
Turn	0.0	24.7	Turn	0.0	26.3
Random	0.0	30.8	Random	0.0	32.8
Total	0.1	100.0	Total	0.1	100.0
RMS Value	12.865		RMS Value	11.722	

Supplementary Fig. 24. The CD spectra. The CD spectra of free Cyt c in water and Cyt c after incubating in DMF (300 μ L) aqueous solution for 6 h (left), and the calculated secondary structure contents of Cyt c based on Yang's reference (right). The volumes of DMF and deionized water were 0.3 mL and 9.0 mL, respectively, which was in agreement with the assembly system. This result indicated that the conformation of enzyme could be well preserved even after 6 h.

GOx in water			GOx in DMF/water		
	Fraction	Ratio		Fraction	Ratio
Helix	0.0	22.2	Helix	0.0	25.1
Beta	0.1	46.0	Beta	0.0	42.2
Turn	0.0	4.9	Turn	0.0	6.5
Random	0.0	26.8	Random	0.0	26.2
Total	0.2	100.0	Total	0.1	100.0
RMS Value	10.956		RMS Value	8.307	

Supplementary Fig. 26. The CD spectra. The CD spectra of free GOx in water and GOx after incubating in DMF (300 μ L) aqueous solution for 6 h (left), and the calculated secondary structure contents of GOX based on Yang's reference (right). The volumes of DMF and deionized water were 0.3 mL and 9.0 mL, respectively, which was in agreement with the assembly system. This result indicated that the conformation of enzyme could be well preserved even after 6 h.

4. The application of HOF crystals in electrochemical immunosensor area is new. I suggest to revise the title to highlight this point.

Response: Thank you for the reviewer's meaningful comment. The title of this manuscript was revised as follow: "Green synthesis of stable hybrid biocatalyst using a hydrogen-bonded, π - π -stacking supramolecular assembly for substrate-sieving biocatalysis and ultrasensitive electrochemical immunosensor."

5. The authors only mention that the method for calculating the detection limit is based on the 3 σ principle, but the specific calculation method should also be clearly stated.

Response: Thank you for the reviewer's comments. The specific calculation method of detection limit has been provided in the Supplementary Method:1.17.

1.17. The calculation of detection limit.

The calculation of detection limit was according to the previous reports (Anal. Chem. 2016, 88, 3203–3210), which was described as follow: DPV measurements for blank samples were executed with three parallel experiments, which exhibited an average photocurrent intensity (I_B) of $-3.289 \mu\text{A}$ with standard deviation (S_B) of 0.085. With signal-to-noise ratio value (k) of 3, the smallest detectable signal (I_L) could be calculated as $I_L = I_B + 3 S_B = -3.035 \mu\text{A}$. Then the value of -3.035 was inserted into the ultralow concentration linear equation ($I = -7.107 \lg c - 8.418$). Thus, the detection limit (c_L) was calculated to be 0.18 pg/mL

6. All figures both in the main text and in Supporting Information should be described and discussed, such as Figure S35 and Figure S36.

Response: We greatly appreciate the reviewer's kind reminder, and we have provided the detail discussion of the zeta potentials (**Supplementary Fig. 44**) and HAADF-STEM images (**Supplementary Fig. 45**).

Supplementary Fig. 44. The Zeta potentials. The Zeta potentials of Au NPs, HRP@HOF-100, HRP@HOF-100-PDDA and Au NPs/HRP@HOF-100. Data are presented as mean \pm s.d. (error bars); $n = 3$.

The zeta potentials of Au NPs and HRP@HOF-100 were measured to be -9.59 mV and -4.92 mV , respectively. It means that Au NPs are hard to combine with HRP@HOF-100 owing to their negative potentials. To solve this problem, we employed the strong cationic polyelectrolyte, poly-(diallyldimethylammonium chloride) (PDDA), which was first combined with the HRP@HOF-100 to obtain the positively charged HRP@HOF-100-PDDA (20.80 mV). After that, the negative Au NPs were easily to anchor to the positive HRP@HOF-100-PDDA through the electrostatic interaction, along with the zeta potential shift from 20.80 mV to 5.98 mV . These results indicated the Au NPs were successfully combined with the HRP@HOF-100.

Supplementary Fig. 45. TEM images. (a) HAADF-STEM image of Au NPs/HRP@HOF-100. (b) The high-resolution TEM image of Au NPs.

The HAADF-STEM image of Au NPs/HRP@HOF-100 showed the existence of Au nanoparticles onto HOF scaffold (**Supplementary Fig. 45a**). In addition, the high-resolution TEM image identified the interplanar spacing of 0.23 nm, ascribed to the (111) lattice plane of Au nanoparticle (**Supplementary Fig. 45b**). These results also demonstrated the successful synthesis of Au NPs/HRP@HOF-100.

We have provided these discussions in the Figure caption of **Supplementary Figs. 44 and 45**.

7. The captions of Figure 1a/Figure S21 should be changed to “PXRD patterns of HRP@HOF-100/Lipase@HOF-100, experimental HOF-100 and simulated HOF-100.”.

Response: We are sorry for this mistake. We have corrected these captions in the main text and Supplementary Information.

8. Abbreviations should be clearly defined when they first appear, including but not limited to: SEM.

Response: According to the reviewer’s kind reminder, we have carefully checked all abbreviations in the main text and Supplementary Information and have been clearly defined them when they first appear.

9. The GC in Figure 4b should be modified to GE, which is in accordance with the text.

Response: We apologize for the negligence, and we have corrected it in **Fig. 5b**.

Fig. 5. Ultrasensitive electrochemical immunosensor. (a) Schematic preparation of Ab2/Au NPs/HRP@HOF-100 electrochemical beacon. (b) The principle of the electrochemical immunosensor based on cyclic biocascade for MUC1 detection. (c) DPV response profile of the developed electrochemical immunosensor as a function of MUC1: 0 (curve a), 1 pg/mL (curve b), 10 pg/mL (curve c), 100 pg/mL (curve d), 1 ng/mL (curve e), 10 ng/mL (curve f) and 100 ng/mL (curve g). (d) The obtained linear regression equation between the currents and the logarithm of different MUC1 concentrations. Data are presented as mean \pm s.d. (error bars); $n = 3$. (e) Selectivity of the electrochemical immunosensor toward diverse samples: blank, MB (100 ng/mL), GOx (100 ng/mL), cholesterol (100 ng/mL), MUC1 (100 pg/mL) and a mixture solution (100 pg/mL MUC1, 100 ng/mL MB, 100 ng/mL GOx and 100 ng/mL cholesterol). Data are presented as mean \pm s.d. (error bars); $n = 3$.

REVIEWERS' COMMENTS

Reviewer #1 (Remarks to the Author):

I have gone through the revised version of the manuscript. The authors have re-highlighted the novelty of this work, especially the differences and advantages over their previous papers. The application area of electrochemical immunosensor is a new trial for enzyme@HOFs materials. Given the novelty, the quality of the work and the high reputation of the group, I would be happy to recommend its acceptance.

Reviewer #2 (Remarks to the Author):

The revised manuscript is well improved, and therefore, is acceptable for publication.

Reviewer #3 (Remarks to the Author):

I think the authors have fully addressed my questions and now the manuscript can be acceptable by Nature Communications.

Response to reviewers' comments:

Reviewer #1 (Remarks to the Author):

I have gone through the revised version of the manuscript. The authors have re-highlighted the novelty of this work, especially the differences and advantages over their previous papers. The application area of electrochemical immunosensor is a new trial for enzyme@HOFs materials. Given the novelty, the quality of the work and the high reputation of the group, I would be happy to recommend its acceptance.

Response: Thank you so much for the positive comments and the acceptance recommendation.

Reviewer #2 (Remarks to the Author):

The revised manuscript is well improved, and therefore, is acceptable for publication.

Response: Many thanks for the positive comments and the acceptance recommendation.

Reviewer #3 (Remarks to the Author):

I think the authors have fully addressed my questions and now the manuscript can be acceptable by Nature Communications.

Response: Many thanks for the positive comments and the acceptance recommendation.